# Emerging Technologies to Extract Fucoxanthin from *Undaria pinnatifida:* Microwave vs. Ultrasound Assisted Extractions

**DOI:** 10.3390/md21050282

**Published:** 2023-04-28

**Authors:** Catarina Lourenço-Lopes, Anxo Carreira-Casais, Maria Carperna, Marta Barral-Martinez, Franklin Chamorro, Cecilia Jiménez-López, Lucia Cassani, Jesus Simal-Gandara, Miguel A. Prieto

**Affiliations:** 1Nutrition and Bromatology Group, Analytical and Food Chemistry Department, Faculty of Food Science and Technology, University of Vigo, Ourense Campus, E-32004 Ourense, Spain; c.lopes@uvigo.es (C.L.-L.); anxocc@uvigo.es (A.C.-C.); mcarpena@uvigo.es (M.C.); marta.barral@uvigo.es (M.B.-M.); franklin.noel.chamorro@uvigo.es (F.C.); mprieto@uvigo.es (M.A.P.); 2Center for Biomedical Research (CINBIO), Neurocircuits Group, Department of Functional Biology and Health Sciences, Campus Universitario Lagoas, Marcosende, Universidade de Vigo, 36310 Vigo, Spain; cecilia.jimenez.lopez@uvigo.es; 3Centro de Investigação de Montanha (CIMO), Instituto Politécnico de Bragança, Campus de Santa Apolónia, 5300-253 Bragança, Portugal

**Keywords:** fucoxanthin, *Undaria pinnatifida*, microwave-assisted extraction, ultrasound-assisted extraction, extraction optimization, response surface methodology

## Abstract

Macroalgae are an extensive resource for the obtention of bioactive compounds, mainly phenolic compounds, phlorotannins, and pigments. Fucoxanthin (Fx) is the most abundant pigment present in brown algae and has shown several useful bioactivities that can be used to fortify products in the food and cosmetic industries. Nevertheless, to date, there is still insufficient literature reporting on the extraction yield of Fx from *U. pinnatifida* species from green technologies. In this regard, the present study aims to optimize the extraction conditions to obtain the highest Fx yield from *U. pinnatifida* through emerging techniques, namely microwave-assisted extraction (MAE) and ultrasound-assisted extraction (UAE). These methods will be compared with the conventional methodologies of heat-assisted extraction (HAE) and Soxhlet-assisted extraction (SAE). According to our results, even though the extraction yield could be slightly higher when using MAE than UAE, the Fx concentration obtained from the alga was double when using UAE. Thus, the Fx ratio in the final extract reached values of 124.39 mg Fx/g E. However, the optimal conditions should also be considered since UAE needed 30 min to perform the extraction, whereas MAE was able to obtain 58.83 mg Fx/g E in only 3 min and 2 bar, meaning less energy expenditure and minimum cost function. To our knowledge, this study obtains the highest concentrations of Fx ever reported (58.83 mg Fx/g E for MAE and 124.39 mg Fx/g E for UAE), with low energy consumption and short times (3.00 min for MAE and 35.16 min for UAE). Any of these results could be selected for further experiments and proposed for industrial scaling-up.

## 1. Introduction

Macroalgae have been used as food sources since ancient times, especially in eastern countries like Japan or China, where they are traditionally consumed. Macroalgae have also been eaten in other parts of the world, such as in Chile, where archaeological remains of their consumption have been found dating back 1400 years [1]. Macroalgae are sources of proteins, minerals, vitamins, pigments, phenolic compounds, and polysaccharides, with multiple health benefits. These molecules have various applications, especially in the food industry, due to their gelling and stabilizing properties. In industrial agriculture, they can be used as fertilizers or as soil decontaminants; and they can also be used for biofuel production [2]. In addition, current projections in the agricultural sector, such as the lack of arable soils or diminishing freshwater resources, make marine organisms of greater interest to the scientific community as a possible source for the production of biofuels, animal feed or other value-added compounds [3]. Nevertheless, the nutraceutical, pharmaceutical, and cosmetic markets still are the most relevant at the economic level [4].

In belonging to the eukaryotic domain, algae are commonly classified by the pigments they produce into three categories: green algae (Chlorophyta), red algae (Rhodophyta), and brown algae (Phaeophyceae) [4]. Among them, brown algae have the highest amount of bioactive compounds, with phlorotannins, pigments and especially fucoxanthin (Fx) standing out [5]. *Undaria pinnatifida* (Harvey), also known as wakame, is a brown seaweed mainly produced in China, Japan, and the Republic of Korea but commercially produced in smaller quantities in France, New Zealand and Spain [6]. It is known as an invasive alga due to its propagation speed and colonization capacity, and as a consequence, it can occasionally cause problems in marine ecosystems. However, *U. pinnatifida* has been approved for human consumption as a non–traditional food substance by the European Union since 1997 [7]. The photosynthetic pigments of this algae are Fx, chlorophylls a, c1 and c2, and some other xanthophylls [4] (Figure 1).

Fx is a secondary metabolite belonging to the carotenoid family found in the chloroplasts of algal cells [5]. This pigment has gathered attention in the last few decades due to its biological properties, such as antioxidant, antitumoral, neuroprotective, anti-obesity, and anti-inflammatory, that make Fx an attractive additive for producing nutraceuticals, cosmetics, or food supplements [8,9,10]. Regarding this ingredient’s safety, Fx extracts from *U. pinnatifida* have been approved for human consumption up to 15 mg per day according to Article 13(1), Regulation (EC) No 1924/2006. Nevertheless, the EFSA has not yet approved the correlation between Fx consumption and body weight regulation claimed by the Fx food supplements [11]. Additionally, Microphyt filed a request for an extract of *Phaeodactylum tricornutum* with a standardized fucoxanthin content (NF 2018/0626 (EFSA 2018c)), which is still being reviewed [7].

Considering macroalgae as an alternative matrix to obtain high-value-added compounds, it is necessary to find and develop efficient and sustainable extraction protocols [3]. In particular, the production of Fx faces different challenges due to its complicated chemical synthesis; thus, future studies on its extraction would favor its commercialization [5]. Although conventional extraction process has been used to obtain Fx, the eco-trends encourage researchers to explore and optimize more respectful and competitive extraction technologies like microwave-assisted extraction (MAE) or ultrasound-assisted extraction (UAE), which allows higher yields while providing a more environmentally friendly approach. Recently, deep eutectic solvents (DESs) have been reported for the extraction of Fx from brown microalgae (*Tisochrysis lutea*) [12] and macroalgae (*Fucus vesiculosus*) [13]. DESs have desirable properties such as thermal stability, adjustable viscosity, polarity, and high solubilization strength to extract compounds like Fx and advantages such as biodegradability, low toxicity and cost, easy production, and being environmentally and ecologically friendly. However, their application is still limited [12,13].

Nevertheless, many studies have been performed to obtain Fx from brown algae through conventional extraction techniques. Maceration extraction (ME) performed with methanol (MeOH) at room temperature (RT) for 24 h was able to recover 9.01 mg/g dw of Fx from *Myagropsis myagroides* [14]; a different one from *Fucus serratus* with hexane/acetone (70:30) at RT, 24 h obtained 3.57 mg/g dw of Fx [15] and two different studies used *U. pinnatifida* to obtain Fx using ethanol (EtOH) at RT, 1 h = 0.7 mg/g dw of Fx [16] or using MeOH at RT, 96 h = 2.67 mg/g dw [17]. Soxhlet-assisted extraction (SAE), similar to ME, has the inconvenience of long extraction time, large amount of solvent and high energy consumption. Still, a study performed using SAE for 12 h obtained 50 µg/mg of Fx from *U. pinnatifida*, using EtOH as solvent at 78 °C [18] or up to 5.5 mg/g dw from *Feldmannia mitchelliae* using ethyl acetate at 80 °C for 16 h [19].

MAE is a green innovative extraction technology that uses a non-toxic procedure to obtain higher yields with less energy expenditure, waste and use of organic solvents [3]. This technique combines solvent extraction with microwave heating power. The energy is transmitted as waves, penetrating the matrix, and interacting with polar molecules, generating heat that increases the kinetics of the extraction. The cell structure is disrupted, and the solute is dissolved into the solvent, diffusing out of the matrix [20,21]. The conversion of electromagnetic energy into calorific energy takes place thanks to two simultaneous mechanisms: ionic conduction and dipole rotation [21].

In addition, MAE has been reported to be effective for the extraction of phenolic compounds and antioxidants from agro-industrial waste [22]. Numerous studies confirm these benefits compared to conventional technologies, as in the case of the extraction of polyphenols from different brown algae (*Carpophyllum flexuosum*, *Dictyota dichotoma*, *Lobophora variegata*, and *Sargassum fluitans*, among others), in which higher yields and shorter extraction times were obtained by MAE when compared to conventional ME with organic solvents. Regarding the extraction of Fx, MAE was employed by another study from *Laminaria japonica*, *Sargassum fusiforme*, and *U. pinnatifida*, but the results were not as high as expected ranging from 0.04, 0.02 and 0.90 mg/g dw of Fx, respectively [23]. Another study compared the performance of MAE and VMAE (vacuum-microwave-assisted extraction) against conventional processes to extract pigments from two marine microalgae (*Cylindrotheca closterium* and *Dunaliella tertiolecta*). The results showed that MAE was the best extraction technique for the *C. closterium* pigments since the use of microwaves accelerated the pigment extraction kinetics, obtaining a higher extraction yield in a few minutes [24].

UAE is another efficient alternative to conventional methods since it can increase the extraction yield of compounds, facilitating the extraction of heat-sensitive compounds. This technique employs ultrasonic waves with frequencies between 20 kHz and 10 MHz, between audible waves and microwave ranges [25]. It is based on the physicochemical principle of acoustic cavitation, that consists of the formation, growth, and collapse of bubbles present in a solvent that is induced by ultrasonic waves. The propagation of ultrasonic waves through the solvent involves the formation of intermittent regions of high and low pressures that generate gas bubbles. These bubbles grow and lead to their compression and rarefaction (expansion), reaching a critical size prior to their collapse [26,27]. Usually, the work temperatures can be lower, thus minimizing the damage to thermolabile compounds. The short extraction time, together with the use of low amounts of solvent, is also a notable advantage of this technique [28].

These advantages can be seen in some studies where ultrasonic treatments with an amplitude of between 20 and 80% at 20 kHz have been used to improve the extraction yield of collagen from marine by-products [29]. In another study, the extraction and purification of polysaccharides from the microalga *Chlorella pyrenoidosa* were performed under 100 °C, extraction solvent of 80% EtOH in water and during 13 min, resulting in a significant yield increase compared to other extraction techniques such as ME [30]. In a UAE-based work, Fx was detected in 0.03 mg/g of dry extract using EtOH as solvent at 25 °C for 3 h from *U. pinnatifida* [6]. Fx was also extracted with UAE from *Padina tetrastromatica* using EtOH to obtain a yield of 0.75 mg/g dw [6].

Table 1 compiles a list of studies that used conventional and innovative techniques to obtain Fx from *U. pinnatifida* and other species. The extraction conditions, detection methods, and obtained yield are also mentioned. Although the most common and efficient technique for obtaining Fx has been ME, it should be noted that different studies have started to appear using green technologies. To date, there is still insufficient literature reporting on the extraction yield of Fx from *U. pinnatifida* species from green technologies such as MAE and UAE. In this regard, the present study aims to optimize the extraction conditions to obtain the highest Fx yield from *U. pinnatifida* through emerging techniques (MAE and UAE) and determine which one obtains better results.

## 2. Results and Discussion

### 2.1. Pigment Identification in Moderate Conditions

The conventional heat-assisted extraction (HAE) was performed with moderate conditions to allow the identification of as many pigments as possible. The applied methodology produced an extraction yield of 38.84% and allowed us to identify 10 compounds in the ethanolic extract. In this study, EtOH was selected due to the following reasons: (i) acetone and EtOH are two solvents used for the extraction of bioactive compounds in the food technology market, (ii) Fx is a polar compound, and (iii) EtOH is more polar than acetone and facilitates the selective heating of the microwaves (for following MAE), reaching higher extraction yields [36].

The chromatographic characterization and quantification of the identified pigment compounds by high-performance liquid chromatography coupled to a diode array detector (HPLC-DAD) are shown in Figure 2. Briefly, four compounds belonging to the chlorophyll family (chlorophyll c2, chlorophyll c1, chlorophyll a, and pheophorbide A), five xanthophylls (Fx, auroxanthin, an Fx derivative, dihydrolutein, and zeaxanthin) and β-carotene as the only representative from the carotene family.

The quantification of all the compounds from the chlorophyll family was determined using the calibration curve of Chl a and obtaining 441.24 µg/g A dw for peak 1 (P1), which corresponds to chlorophyll c2, 303.36 µg/g A dw for peak 2 (P2) which corresponds to chlorophyll c1, 29.45 µg/g A dw for peak 8 (P8) corresponding to chlorophyll a, and 23.37 µg/g A dw for peak 9 (P9) corresponding to pheophorbide A. The xanthophylls were quantified using the Fx calibration curve and peak 3 (P3), which corresponds to Fx, and obtained 2254.28 µg/g A dw. Peak 4 (P4) corresponded to auroxanthin (14.25 µg/g A dw), peak 5 (P5) corresponded to Fx derivative (378.43 µg/g A dw), peak 6 (P6), corresponded to dihydrolutein (1.84 µg/g A dw) and peak 7 (P7) corresponded to zeaxanthin (47.97 µg/g A dw). At last, β-carotene was identified as peak 10 (P10) and quantified according to its calibration curve, obtaining 2484.85 µg/g A dw. This information is summarized in Figure 2**.**

From all the identified pigments, Fx and β-carotene stood out due to the easily obtained higher yields. There are currently several industries specialized in the extraction of β-carotene or even chlorophylls from many different vegetable matrixes, but the Fx extraction is only present in marine life and still quite unexplored. There are several studies extracting Fx through HAE reported in the literature from brown algae. Most of them use MeOH as the extraction solvent reaching values up to 4.96 mg/g from fresh *U. pinnatifida* [37], 6.42 mg/g dw from *Dictyota dicotoma* [14], 3.70 mg/g dw from *Sargassum horneri* and 1.80 mg/g dw from *Sargassum thunbergia* [32]. However, MeOH shows more concerns in terms of applicable legislation, as it is toxic compared to EtOH. Therefore, we decided to focus this optimization study on this pigment and EtOH as a solvent [38].

### 2.2. SAE as Reference Method of Extraction

As previously stated, SAE has been used as a reference method for the extraction of pigments. This method was selected based on previous work that showed Fx was not so thermolabile and could withstand temperatures up to 85 °C for long periods of time (even higher than 60 min) [39]. Fx has been extracted from *U. pinnatifida* using EtOH as a solvent and obtaining 0.05 mg/g dw [18], using ethyl acetate from the brown alga *Feldmannia mitchelliae* and *Sargassum swartzii* C. Agardh, obtaining 5.50 mg/g dw and 0.17 mg/g dw, respectively [34].

In this study, Fx was subjected to 78.4 °C (EtOH ebullition temperature) and resisted degradation during 4 h extraction. The extraction was performed in duplicate with two selected time periods (2 and 4 h). Although the extraction yield was almost unchanged, between 168.30 and 168.83 mg E/g A dw, some differences were found regarding the specific pigment content of the obtained extracts, as described below:Regarding the extraction of Fx, the content slightly differed between the two tested extractions. The 2 h extraction obtained 3.68 mg Fx/g A dw and 21.90 mg Fx/g E dw, whereas the 4 h extraction obtained higher values of 4.58 mg Fx/g A dw and 30.80 mg Fx/g E dw. These results are in agreement with previously published data regarding Fx thermal resistance [39]. Previous studies with SAE obtained much lower yields with 0.05 mg/g dw from *U. pinnatifida* using EtOH as a solvent for 12 h [18]. Other brown algae obtained better results with different solvents, as seen in Table 1 and Figure 2. When using ethyl acetate at 80 °C, *Feldmannia mitchelliae* obtained 5.50 mg/g dw in 16 h [19], and *Sargassum swartzii* C. Agardh obtained 0.17 mg/g dw in 6 h [34]. Using n-hexane, 0.45 mg/g dw of Fx was obtained from *Saccharina japonica* at 40 °C for 16 h [33].For chlorophyll a, the difference was even smaller, obtaining 1.00 mg Chl/g A dw and 45.60 mg chl/g E dw in 2 h and 1.26 mg Chl/g A dw and 45.89 mg Chl/g E dw in 4 h. The difference obtained is probably not relevant to justify doubling the time and the energy spent on the extraction, especially when scaling up this process.For β-carotene, the obtained results were 0.24 mg β-car/g A dw and 5.22 mg β-car/g E at the 2 h and 0.22 mg β-car/g A dw and 4.31 mg β-car/g E dw at 4 h extraction. These last results are the only ones where the 4-h extraction performed slightly worse, suggesting that β-carotene might be the least thermos-resistant pigment of the three in the study, which resulted in its degradation, leading to lower yields.

Nevertheless, the SAE obtained up to 4.58 mg Fx/g A dw when compared to HAE, 2.25 mg Fx/g A dw and 1.26 mg Chl/g A dw as opposed to 0.23 mg Chl/g A dw, respectively. For the β-carotene, we can observe once again that the SAE conditions were not appropriate, as the yield decreased from 2.48 mg/g A dw to only 0.24 mg/g.

### 2.3. Optimization by MAE and UAE

#### 2.3.1. Variable Selection for the Experimental Design

Temperature and time of extraction are very variable. According to previous research, temperatures for Fx extraction ranged from 4 °C up to 65 °C, while evaluated times ranged from 15 min up to 96 h. The extraction *time* (*X*_1_) was selected as a variable and studied from 5 to 55 min for UAE and 3 to 23 min for MAE based on previous research. The ranges for *power* or *pressure* (*X*_2_) were set from 100 to 500 W for UAE and from 2 to 12 bar for MAE), according to the specifications of the equipment and preliminary results in our laboratory. For the extraction solvent to obtain Fx, the most utilized solvents are MeOH, EtOH, and acetone, which have been applied at different percentages. Other studies have been performed using alternative options such as water, hexane, chloroform, dichloromethane, heptane, or diethyl ether. According to some investigations, the preferred solvent was EtOH [40,41]. However, in a two-level full factorial design for the extraction of Fx from *Sargassum siliquosum* and *S. polycystum*, the best solvent was MeOH [42]. Similarly, another study based on nine different brown algae species determined acetone as the best solvent for the extraction of Fx. [43]. From our previous results, EtOH was selected and the studied *ethanol percentage* (*X*_3_) from 35 to 100% for UAE; and 20 to 100% for MAE was chosen.

#### 2.3.2. Experimental Data for All Response Criteria from CCCD and Theoretical RSM Analysis

After selecting the target pigment, the optimization of Fx extraction from *U. pinnatifida* was performed using two innovative methodologies: MAE and UAE. These two methods have been proven to improve the extraction of compounds from vegetable matrixes and Fx from other brown algae. Furthermore, these two methodologies are green extraction techniques that require less solvent and short extraction times which translates into less energy spent. Studies performed using MAE and *U. pinnatifida* obtained 0.73 and 0.90 mg/g dw of Fx [23], but these results could be improved after optimizing the extraction process. Regarding UAE, 0.03 mg/g dw was obtained in a methanolic extraction of *U. pinnatifida,* but better results were obtained in a study that used EtOH, with *Padina tetrastromatica* obtaining 0.75 mg/g dw.

The response surface methodology (RSM) allows the evaluation of the effects of a set of variables and the interactions between them. The *Circumscribed Central Composite Design (CCCD)* has been applied by a number of researchers in the optimization of multiple food processing methods [44]. In this case, a design with five levels of variation for the three independent variables: *time* (*t* or *X*_1_), *power* (*Pw* or *X*_2_) or *pressure* (*P* or *X*_2_), and *ethanol concentration* (*S* or *X*_3_), was applied. A detailed description of the coded and natural values of the selected variables for each extraction method in the *CCCD* design and the obtained responses is presented in Table 2.

From Table 2, higher yields of extraction (*Y*_1_) were obtained in the case of MAE, ranging from 13 to 56%, whereas 5–52% for UAE. For MAE, the highest result was obtained for experimental run number 17, using shorter times, maximum energy, and a lesser percentage of EtOH concentration. For UAE, run number 13 obtained the highest yield using medium time and energy and a low percentage of EtOH concentration. However, for the concentration of Fx (*Y*_2_), higher extraction yields were obtained for UAE than MAE, corresponding up to 20.89 and 10.18 mg Fx/g A, respectively. This difference might be related to the energy applied by each technique. MAE obtained slightly higher yields (in terms of dw), while UAE was able to selectively extract more Fx. Considering the *Y*_2_/*Y*_1_ ratio, it was maximum for the experimental run number 16 for MAE, using shorter time, minimum energy, and the highest EtOH concentration obtaining 67.27 mg Fx/g E. For UAE, run number 20, with minimum energy, the longest time, and the highest EtOH concentration, obtained 161.48 mg Fx/g E. This first approximation, without fitting any model, already gives an idea about the efficiency of UAE over MAE.

The obtained responses (*Y*_1_, *Y*_2_ and *Y*_2_/*Y*_1_) were fitted to Equation (8) using non-linear least squares estimates to obtain a mathematical expression that allows to make predictions and extrapolate the optimal conditions that maximize the extraction of Fx, simplifying the complexity of the model. Those parametric values were obtained and are presented in Table 3.

The coefficients that showed effects higher than the confidence interval of the parameter (α = 0.05) were considered as not significant (ns) and not considered for the development of the model. These coefficients were used to build the third-order non-linear Equations (1)–(6) for each extraction technique (MAE and UAE).
(1)$$Y_{1}^{MAE}=49.87+2.61P-3.87t^{2}-4.27S^{2}+1.08t^{3}-3.73S^{3}+0.48tPS+0.39t^{2}P^{2}S^{2}$$
(2)$$Y_{2}^{MAE}=4.26-0.31t-1.38P+2.79S-0.69P^{2}-0.79S^{2}-0.56S^{3}-0.77PS$$
(3)$${Y_{2}/Y_{1}}^{MAE}=11.00-2.09t-6.54P+6.54S-5.20PS+0.95P^{2}S^{2}$$
(4)$$Y_{1}^{UAE}=45.41-4.47t+15.24Pw-6.49t^{2}-10.01{Pw}^{2}-4.88S^{2}+1.79t^{3}-3.86{Pw}^{3}-3.04S^{3}-1.03tPw\;-1.61PwS+0.70tPwS-6.26t^{2}{Pw}^{2}+3.85t^{2}{Pw}^{2}S^{2}$$
(5)$$Y_{2}^{UAE}=18.49+4.76Pw+3.40S-2.48t^{2}-3.03{Pw}^{2}-4.10S^{2}-1.09{Pw}^{3}+0.55tS+0.68PwS-4.91t^{2}{Pw}^{2}\;+2.39t^{2}{Pw}^{2}S^{2}$$
(6)$${Y_{2}/Y_{1}}^{UAE}=39.12-5.36t^{2}+8.05S^{2}-13.01S^{3}+2.21tS+0.95t^{2}{Pw}^{2}S^{2}$$

Each parametric value showed the linear, quadratic, cubic and interactive (both linear and quadratic) effects: the absolute value corresponds to the weight of the variable in the equation, and the sign (positive or negative) indicates the response performance. For example, in terms of $Y_{1}^{MAE}$, a positive linear effect was caused by *P*, and negative quadratic effects were observed for *t* and *S*. A positive cubic was observed for *t,* whereas *S* showed a negative cubic effect. Both positive linear and quadratic interactions were found between *t*, *P*, and *S*. In terms of the statistical analysis, the quadratic regression model resulted in the determination of the *R^2^* coefficient (Table 3). All responses for MAE and UAE showed values higher than 0.85. Thus, the model explains more than 85% of the variability. Higher values were obtained for UAE compared to MAE, suggesting a better experimental design with less variability.

#### 2.3.3. Response Patterns and Optimal Conditions

The response patterns derived from the parametric values previously explained can also be expressed in a graphical representation by using 3D plots. Figure 3 shows an example of the response surface and contour plots based on the model equation and how the analysis is developed.

In this case, it is shown the graphic representation of the concentration of Fx (*Y_2_, mg Fx/g A*) in UAE. First, the graphical analysis of the response is carried out to represent the response values in net surface plots as well as in contour plots (Figure 3A). Both types of graphs show the same information, only differing in the dimension, 3D or 2D, respectively. The net surfaces were predicted using the third-order polynomial model of Equation (8), and in particular, the model equation corresponds to Equation (5). Then, in Figure 3B, the statistics results are represented. The quadratic regression model compares the experimentally observed data to those predicted values, showing only an error of less than 10% (*R^2^* = 0.9034). Next to this graph, the distribution of the residual values (%) for each variable is depicted. In these graphs, it is determined that the largest error does not reach 6%, and the majority are within 3%, strengthening the reliability of the mathematical model. This evaluation makes it possible to confirm no autocorrelations or patterns. The values obtained for the three variables are random.

Finally, Figure 3C shows how the optimal results behave for each of the variables. Thus, the left graph shows how *t* behaves in the optimum conditions of *Pw* and *S*. The same is shown for *Pw* and *S* in the next two graphs. Considering the results shown by this analysis, it is possible to study the economic profitability of a process, allowing interaction with the model, seeing how the optimal extraction values behave and adapting them to lower the costs of the extraction process. For example, in this case $Y_{1}^{MAE}$, the analysis of the variable’s behavior shows that better results are obtained working between the factorial and axial points of the experimental design.

In the same way as shown in Figure 3A,B, Figure 4 shows the graphical profile patterns derived from the parametric values of these mathematical models in terms of response values (*Y*_1_, *Y*_2_, and *Y*_2_/*Y*_1_) for MAE and UAE. In Figure 4A the binary actions between the variables (Equations (1)–(6)) are displayed when the excluded variable is positioned at the center of the experimental domain. From these plots, some approximations can be assumed. For example, S seems to dominantly influence the responses compared to other variables. Figure 4B illustrates the capability to predict the obtained results and the residual distribution as a function of each of the considered variables.

The goodness of fit of the model is illustrated by the ability to simulate response changes between the observed and predicted data and the residual distribution as a function of each variable. Consequently, the distribution of the residue confirms reliability higher than 85% in all cases (*R^2^* = 0.85), as shown in Table 3.

Based on the experimental and statistical analysis (Table 3), numerical optimizations were performed to establish the optimum values of the independent variables to obtain desirable response levels and maximize the efficiency of each extraction technique. The optimal values for each of the responses were defined as follows and presented in Table 3B. For Fx extraction from *U. pinnatifida* using MAE:
For the extraction yield ( $Y_{1}^{MAE}$
), the optimal variable conditions were 10.27 min, 12.00 bar and 20% of EtOH to obtain a response of 60.25%.For the Fx concentration ( $Y_{2}^{MAE}$
), the optimal variable conditions were 3.00 min, 2.00 bar and 100% of EtOH to obtain a response of 10.01 mg Fx/g A.For the Fx concentration ratio in the extract ( ${Y_{2}/Y_{1}}^{MAE}$
), the optimal variable conditions were 3.00 min, 2.00 bar and 100% of EtOH to obtain a response of 58.83 mg Fx/g E.

For Fx extraction from *U. pinnatifida* using UAE:
For the extraction yield ($Y_{1}^{UAE}$
), the optimal variable conditions were 21.63 min, 388.68 W and 35% of EtOH to obtain a response of 54.13%.For the Fx concentration ( $Y_{2}^{UAE}$
), the optimal variable conditions were 30.46 min, 374.46 W and 76.55% of EtOH to obtain a response of 20.91 mg Fx/g A.For the Fx concentration ratio in the extract ( ${Y_{2}/Y_{1}}^{UAE}$
), the optimal variable conditions were 35.16 min, 300.00 W and 100% of EtOH to obtain a response of 124.39 mg Fx/g E.


According to these results, even though the extraction yield could be slightly higher using MAE than UAE, Fx concentration in the alga was double using UAE, and thus, the Fx ratio in the final extract reached values of 124.39 mg Fx/g E. However, the optimal conditions should also be considered since UAE needed 30 min to perform the extraction, whereas MAE was able to obtain 58.83 mg Fx/g E in only 3 min and 2 bar, meaning less energy expenditure and minimum cost function.

The highest reported concentration of Fx from *U. pinnatifida* was 2.671 mg/g dw and 4.96 mg/g fresh weight using the maceration technique [17,37]. Nevertheless, Fx has been previously extracted from brown algae using innovative extraction techniques. MAE was applied to obtain Fx *L. japonica*, *U. pinnatifida*, and *S. fusiforme* using EtOH and acetone as solvent at 50 °C for 10 min and obtained concentrations ranging from 2 to 110 mg/100 g [23]. In another study, Fx was extracted from *L. japonica* and *S. fusiforme*, but a higher yield was obtained using maceration than MAE [23,45,46]. There is even a patent (ref. no. CN104327017A) that applies MAE for 5 min using EtOH and, later, a two-step separation process to isolate Fx from other components. From this point of view, the proposed optimal conditions ameliorate this extraction process and shorten it to only 3 min. On the other hand, UAE has also been applied to extract Fx. For example, *Padina tetrastromatica* was submitted to extraction under EtOH 80%, 50 °C, and 30 min to obtain 750 µg/g dw of Fx. In this case, the yield was better than using conventional extraction techniques [35,47]. Fx was also extracted using other alternative techniques to obtain up to 6.42 mg/g dw of Fx [5].

Considering previous research, conventional techniques represent a low-cost alternative with lower performance difficulty and usually obtained better extraction yields for Fx [48]. However, longer extraction times and environmental applications, mostly due to solvent toxicity, have prompted the utilization of non-conventional techniques [5]. UAE and MAE are more environmentally friendly, although scaling up can be a limiting step to applying these techniques. To our knowledge, this is the study where the highest concentrations of Fx have been obtained (58.83 mg Fx/g E for MAE and 124.39 mg Fx/g E for UAE), with low energy consumption and short times (3 min for MAE and 35.16 min for UAE). Any of them could be selected for further experiments and proposed industrial scaling-up. Mathematical equations were calculated based on limited diapasons (Table 2) to predict the extraction yield and fucoxanthin concentration that can be combined with other equations defined by the industry that consider the economic and environmental aspects, among others.

## 3. Materials and Methods

### 3.1. Samples Collection

All the experiments were carried out using *U. pinnatifida*, a brown alga from the Phaeophyceae family. The alga was manually collected from Galician coastlines in December 2019 (mature sporophytes) and provided by the company Algamar. The reproductive phase and season should be considered when analyzing and discussing the obtained data due to their influence on the yield of active compounds [49]. The fresh alga was washed with tap water to remove some salts and other macroscopic impurities, stored in plastic zipper bags and frozen at −80 °C until it was lyophilized (LyoAlfa 10/15 from Telstar, University of Vigo). Then, the alga was converted into powder (~20 mesh) and stored in falcon tubes at −20 °C until used.

### 3.2. Conventional Heat-Assisted Extraction for Pigment Identification

The heat-assisted extraction (HAE) was performed using 0.6 g of lyophilized alga powder and 20 mL of EtOH for a solid/liquid (*S/L*) ratio of 30 g/L, with a similar protocol used before [36]. The mixture was incubated at 50 °C in an orbital shaker at 150 rpm for 24 h, protected from light. After that, the supernatant was removed, and the pellet was re-extracted twice with 10 mL of EtOH for 1 h more. Then, the total volume was centrifuged at 4800 rpm for 8 min. From the final extract, 5 mL was used to calculate the dry weight (dw) (at 104 °C for 24 h), and the remaining extract was evaporated in a rotary evaporator at 40 °C to obtain a dry extract. The dry extract was then resuspended in 10 mL of 80% aqueous EtOH, filtered with a syringe filter (0.22 µm pore size) into amber vials and analyzed through HPLC-DAD.

### 3.3. Soxhlet-Assisted Extraction as Reference Method

The Soxhlet-assisted extraction (SAE) was carried out using a Buchi Extraction System B-811 (Switzerland). An amount of 2 g from *U. pinnatifida* was added to a cellulose cartridge and carefully closed. The cellulose cartridge was introduced in the intermediate body of the Soxhlet extractor and closed. Then, 100 mL of EtOH was added to the lower camera and attached to the apparatus body. The extraction was performed by setting the parameters for 2 and 4 h, with the program for EtOH as the solvent, at 78.4 °C. This corresponds approximately to a cycle every 7 min, meaning 17 cycles in 2 h and 34 cycles in 4 h. At the end of the extraction time, the extracts were filtered using a syringe nylon filter with 0.22 µm in diameter into amber vials. The vials were stored in a freezer at −80 °C until their analysis in HPLC-DAD. Each experimental point was carried out in duplicate, expressed in terms of mean ± standard deviation (SD) and expressed in terms of dry weight (dw).

### 3.4. Chemical Analysis through HPLC-DAD

The pigments’ content of the extract obtained by HAE was determined by a Waters HPLC equipment [including Waters 600 controller and Waters 600 pump, Waters 2996 photodiode array detector (DAD) (1.2 nm optical resolution), Waters 717 plus autosampler, and an AF in-line degasser from Waters]. A Waters Nova-Pak C18 column (150 × 3.9 mm, 4 μm particle, WAT 088344) was used and stabilized at 25 °C. The mobile phases were prepared with HPLC-grade solvents, and the eluents used were A: 5 mM ammonium acetate in milli-Q water, B: 5 mM ammonium acetate in MeOH, and C: pure ethyl acetate. The eluent gradient started with 5% of A and 95% of B during 8 min, then changed to 50% B and 50% C until min 20; up to min 35, the eluents were 50% A and 50% B and the run ended at min 40 with 30% A and 70% B. The flow rate was set at 0.5 mL/min, and the injection volume was 50 µL. The pigments were detected using a DAD with absorbances between 450 nm and 700 nm [50]. The quantification of all chlorophylls was determined according to the calibration curve of chlorophyll a standard (y = 3.35 × 10^8^ x − 3.86 × 10^4^, *R^2^* = 0.9993; LOD = 0.65 µg/mL, LOQ = 1.96 µg/mL); the quantification of all xanthophylls was determined according with the calibration curve of Fx standard (y = 5.10 × 10^1^ x + 2.76 × 10^−2^, *R^2^* = 0.9999; LOD = 0.01 mg/mL, LOQ = 0.04 mg/mL); and the quantification of carotenes was determined according with the calibration curve of β-carotene standard (y = 7.84 × 10^5^ x + 2.72 × 10^4^, *R^2^* = 0.9904; LOD = 0.01 mg/mL, LOQ = 0.02 mg/mL). The three standards were purchased from Sigma.

### 3.5. Extraction Yield

The extraction yield of all extractions was measured in terms of the dry weight (dw). In order to do so, 30 mL crucibles were placed at 104 °C for 1–2 h in a TCF 120 Forced air Oven (Argo Lab). Crucibles were then cooled down for 15–20 min in a desiccator right before gravimetrically measuring their weight. Later, 5 mL of the extracted solution was added into the crucibles and put back in the oven for 24 h. After that time, crucibles were cooled down in the desiccator and weighed. The extraction yield was then calculated following Equation (7):(7)$$Yield\left(\%\right)=\frac{P_{24h}-P_{0h}}{\left(\frac{M_{dw}\times V_{crucible}}{V_{solvent}}\right)\times \frac{\left(100-H_{alga}\right)}{100}}\times 100$$
where *P_0h_* is the weight of the crucible; *P_24h_* is the weight of the crucible after 24 h of drying; *M_dw_* is the mass of lyophilized alga used to perform the extraction; *V_crucible_* is the volume of extracted solution added to the crucible (5 mL); *V_solvent_* is the volume of solvent used to perform the extraction (20 mL); *H_alga_* is the humidity (%) of each algae species.

### 3.6. Optimization of the Extraction Process from Undaria Pinnatifida

#### 3.6.1. Microwave-Assisted Extraction (MAE)

For microwave-assisted extraction (MAE), a multiwave-3000 microwave extraction system (Anton-Paar, Ostfildern, Germany) was employed. The system consisted of a closed extraction chamber with 16 vessels equipped with an infrared sensor, a pressure and temperature sensor (P/T), a vessel detection sensor, and a magnetic stirrer at the base. The three variables and ranges in the study were *time* (*t* or *X*_1_, 3 to 23 min), *pressure* (*P* or *X*_2_, 2 to 12 bar) and *ethanol concentration* (*S* or *X*_3_, 20 to 100%). Sample preparation consisted of weighing 0.6 g of lyophilized *U. pinnatifida* in each equipment container and then adding 20 mL of solvent (*S/L* ratio of 30 g/L), placing a magnetic stirrer inside the containers, and closing them. The potency of the equipment was set at its maximum value of 1400 W. After the extraction was completed, the samples were rapidly placed in an ice bath for 5 min to avoid the degradation of thermolabile compounds. Subsequently, the samples were centrifuged at 9000 rpm for 15 min [51]. The supernatant was paper filtered and then filtered to amber using a syringe nylon filter of 0.22 µm. The vials were stored in a freezer at −80 °C until their later analysis in HPLC-DAD. Each experimental point was expressed in terms of dry weight (dw).

#### 3.6.2. Ultrasound-Assisted Extraction (UAE)

The ultrasound-assisted extraction (UAE) was carried out in an ultrasonic device (Optic Ivymen System sonicators, model CY-500, Spain). The algae sample (1.05 g) was extracted in 35 mL of solvent, thus maintaining the same *S/L* ratio of 30 g/L. In this case, the variables and their ranges were *time* (*t* or *X*_1_, from 5 to 55 min), *power* (*Pw* or *X*_2_, from 100 to 500 W), and *ethanol concentration* (*S* or *X*_3_, from 35 to 100%), while the temperature was controlled to be kept below 30–35 °C, thanks to an ice bath [52]. The obtained extracts were centrifuged at 8400 rpm for 7 min to eliminate any impurities. The supernatant was filtered through a syringe nylon filter with 0.22 µm to amber vials and kept at −80 °C until further analysis in HPLC-DAD. Each experimental point was expressed in terms of dry weight (dw).

#### 3.6.3. Experimental Design, Analysis Model and Statistical Evaluation

To design the optimization experiment, a series of single-variable experiments and previous work was performed. Then, the most relevant variables for each extraction technique were selected along with their appropriate ranges [53]. In order to obtain the conditions that would maximize the yield, a Response Surface Methodology (RSM) was applied along with a Circumscribed Central Composite Design (CCCD). The effects of the three defined variables for each extraction technique were studied by using a CCCD with five levels, which generated 28 combinations of responses, to obtain a better predictive capacity of the model (Table 2) [54]. The responses used in the optimization process were three: *Y*_1_ (%), which represents the extraction yield; *Y*_2_ (mg Fx/g A), which represents the total content of Fx in the algae dw; and *Y_2_/Y_1_* (mg Fx/g E), which represents the purity of the extracts obtained.

##### Mathematical Model

In order to determine the optimum conditions for each extraction technique, a mathematical model was implemented, allowing the maximization of the extraction yield and Fx concentration. The RSM models were fitted by calculating least-squares using the third-order polynomial model from Equation (8):(8)$$Y=b_{0}+\sum_{i=1}^{n}b_{i}X_{i}+\sum_{\begin{array}{c} i=1\\ j>i \end{array}}^{n-1}\sum_{j=2}^{n}b_{ij}X_{i}X_{j}+\sum_{i=1}^{n}b_{iijj}X_{i}^{2}X_{j}^{2}\sum_{i=1}^{n}b_{iii}X_{i}^{3}$$
where *Y* is the dependent variable (response variable); *X_i_* and *X_j_* are independent variables; *b_0_* is the constant coefficient; *b_i_* is the linear effect coefficient; *b_ij_* is the linear interactive effect between two variables coefficient; *b_ii_* is the quadratic effect of each variable coefficient; *b_iijj_* is the quadratic interactive effect between two variables coefficient; *b_iii_* is the cubic effect of each variable coefficient; and *n* is the number of variables.

##### Procedure for Optimization of Variables

The experimental trials were randomized to minimize unpredictable effects on the observed responses. In order to maximize the responses produced by the model, a *simplex* method tool was used to solve non-linear problems. Coded values were limited to avoid unnatural conditions (e.g., t < 0).

##### Numerical Methods, Statistical Analysis, and Graphic Illustrations

All statistical calculations, fitting procedures, and coefficient estimates were performed using a Microsoft Excel spreadsheet. The graphic illustrations were created in DeltaGraph v.7. from the obtained data. The statistical analysis of the experimental results was carried out in four phases:Determination of the coefficients: the parametric estimates were obtained by minimizing the sum of the quadratic differences between the obtained and predicted values, using the least squares method (quasi-Newton) by the “Solver” macro in Microsoft Excel, which allowed the rapid analysis of a hypothesis and its consequences.Significance of the coefficients: to obtain significance values, the confidence intervals of the parameters were calculated using “*SolverAid*”. The model was simplified by discarding the non-statistically significant terms for the *p*-value (*p* > 0.05).Consistency of the model: it was carried out through Fisher’s F test (α = 0.05) to determine if the constructed models were adequate to describe the data obtained.Other statistical evaluation criteria: to re-verify the uniformity of the model, the following criteria were applied: “*SolverStat*” to evaluate the prediction uncertainties of parameters and models, as well as the R² value, interpreted as the proportion of versatility of each dependent variable explained by the model.

## 4. Conclusions

Macroalgae are valuable marine resources due to their diverse secondary metabolites like phenolic compounds and pigments for developing new ingredients or fortifying products in the food and cosmetic industries. Fucoxanthin (Fx) is one of the major carotenoids found in brown algae, and its structure is responsible for their diverse properties, like strong antioxidant, anti-cancer, anti-inflammatory and anti-obesity activities, among others. The optimization of the extractive process of this molecule is of great interest to the industry due to the several therapeutic activities described.

SAE, the reference method for the extraction of Fx, obtained an extraction yield of up to 168.83 mg E/g A dw. The 2 h extraction obtained 3.68 mg Fx/g A dw and 21.90 mg Fx/g E dw, whereas the 4 h extraction obtained higher values of 4.58 mg Fx/g A dw and 30.80 mg Fx/g E dw. After the optimization of the UAE and MAE extraction, we can conclude that even though the yield was a bit higher for MAE (60.25% against 54.13%), UAE was able to obtain higher concentrations of Fx (20.91 mg Fx/g A for UAE and 10.01 mg Fx/g A for MAE) and increase the purity of the extracts (124.39 mg Fx/g E for UAE and 58.83 mg Fx/g E for MAE). In general, MAE obtained better results for shorter times, low pressure, and higher concentration of ethanol, whereas UAE needed medium times, medium power, and higher ethanol concentration. These results are far higher than the ones previously described in the literature and higher than the ones obtained with the conventional SAE. Both methodologies have the potential for industrial scaling up, allowing for the obtention of higher yields and minimizing the solvent and energy used, thanks to their short extraction times. The study limitations are related to the selected ranges of the studied variables and the industrial scaling requirements of these techniques. Further research should focus on the isolation and purification of Fx after extraction.

## Figures and Tables

**Figure 1 marinedrugs-21-00282-f001:**
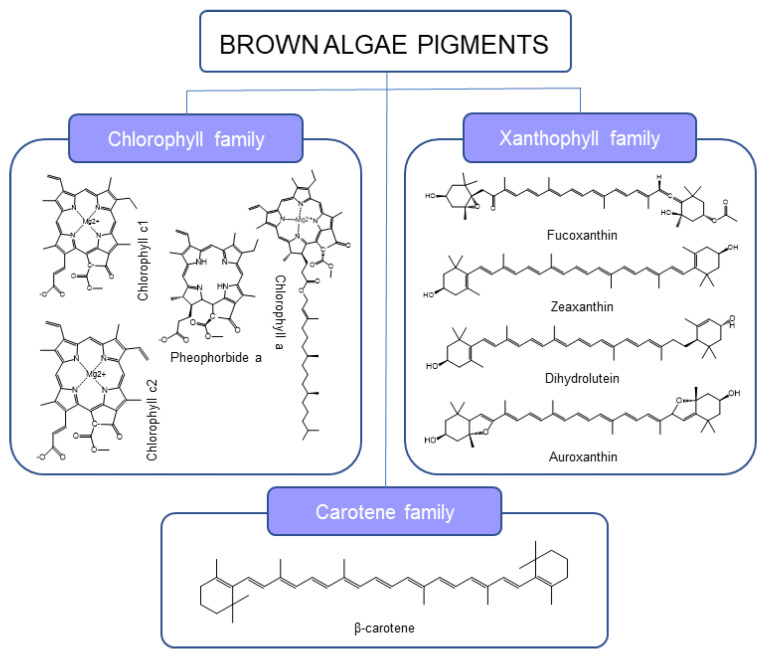
Pigment families and main molecules identified on *U. pinnatifida* ethanolic HAE extracts.

**Figure 2 marinedrugs-21-00282-f002:**
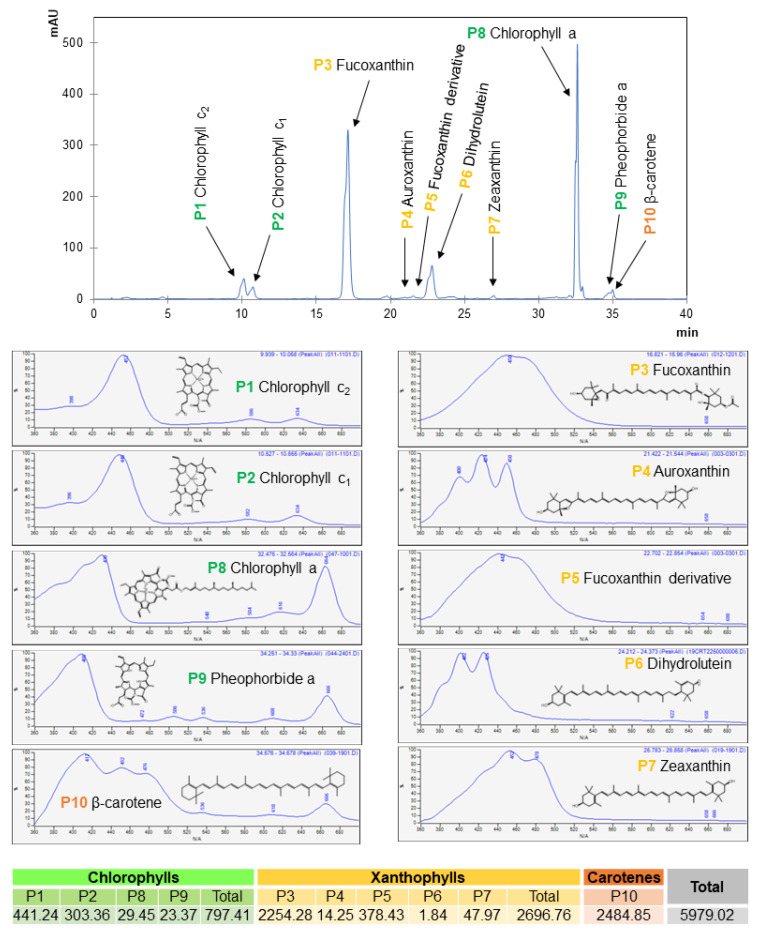
Pigment identification of *U. pinnatifida* extracted with EtOH through HAE extraction and analyzed with HPLC-DAD. The extraction yield obtained was 38.84%, and the pigment content is expressed in µg/g A dw.

**Figure 3 marinedrugs-21-00282-f003:**
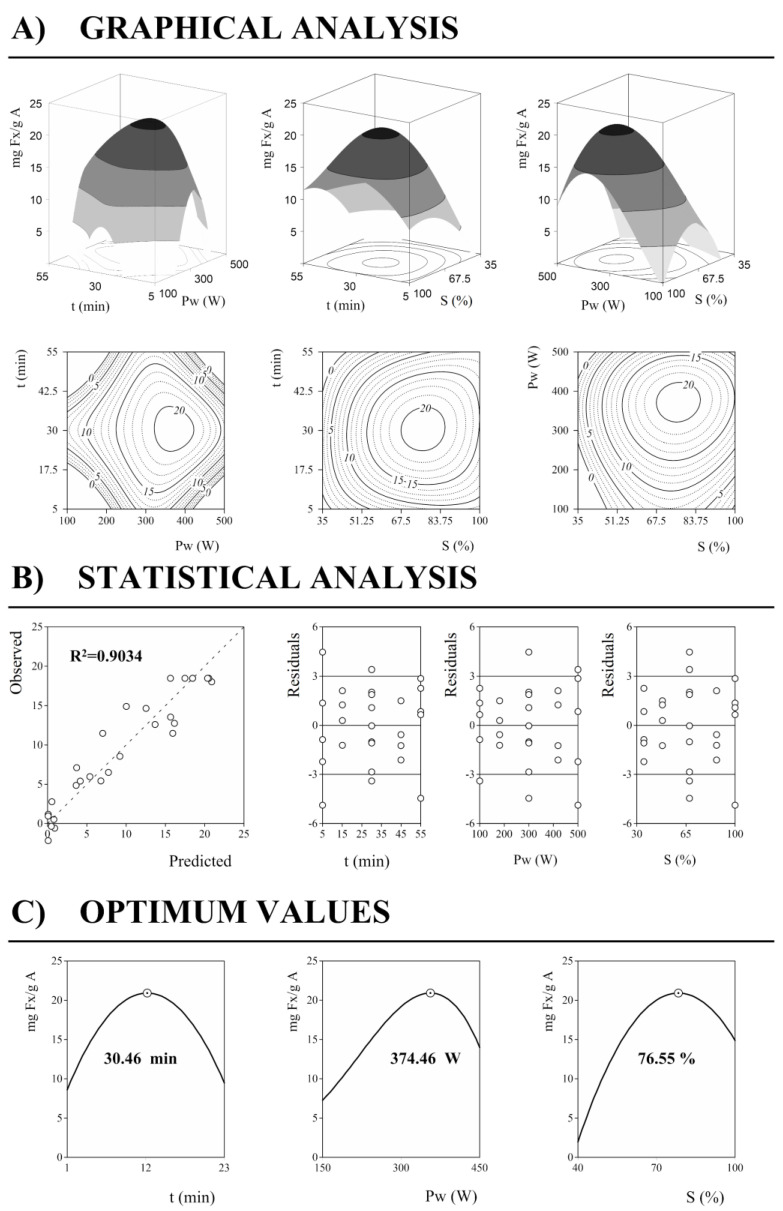
Graphic representation of UAE results as a function of the three main variables involved (*X*_1_, *X*_2_, and *X*_3_) for the concentration of Fx (*Y_2_, mg Fx/g A*). (**A**) Graphical analysis by net surfaces that represents the 3D response surface predicted with the third order polynomial of Equation (8). (**B**): Quadratic regression model and the residual distribution as a function of each of the variables. (**C**): Individual 2D responses of the assessed variables (*t, Pw, S*). The variables in each of the 2D graphs were positioned at the individual optimal values of the others (Table 3). The dots (⊙) presented alongside each line highlight the location of the optimum value. Lines and dots are generated by the theoretical third-order polynomial (Table 3).

**Figure 4 marinedrugs-21-00282-f004:**
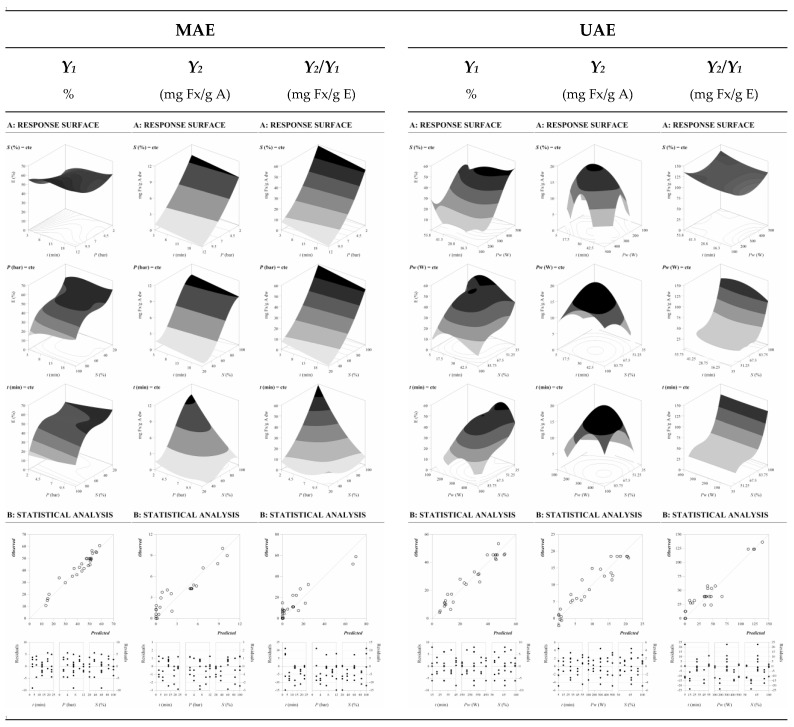
Graphic representation of MAE and UAE results as a function of the three main variables involved (*X*_1_, *X*_2_, and *X*_3_) for all the studied responses of Fx concentration (*Y*_1_, *Y*_2_, and *Y*_2_/*Y*_1_) from *Undaria pinnatifida* (UP). (**A**): Graphical analysis by net surfaces that represents the 3D response surface predicted with the third order polynomial of Equation (8). The binary actions between variables are presented when the excluded variable is positioned at the individual optimum (Table 3). (**B**): Quadratic regression model of the responses between the predicted and observed data; and the second one, the residual distribution as a function of each of the variables. The statistical design and results are described in Table 3.

**Table 1 marinedrugs-21-00282-t001:** Brown macroalgae as a source of Fx extracted by heat-assisted extraction (HAE), Soxhlet-assisted extraction (SAE), microwave-assisted extraction (MAE) and ultrasound-assisted extraction (UAE) techniques, detection methods used to determine its presence and quantification values (Fx concentration is expressed as mg Fx/g dw).

Technique	Solvent		Conditions	Method	Fx	Ref.
** *Undaria pinnatifida* **						
**HAE**	EtOH		RT, 1 h	HPLC-DAD	0.70	[16]
					0.07 *	
	MeOH				2.08	[31]
					4.96 *	
			RT, 96 h	HPLC-DAD, ^1^H NMR and ^13^C NMR	2.67	[17]
**SAE**	EtOH		78 °C, 12 h	HPLC-UV	0.05	[18]
**MAE**	EtOH		60 °C, 10 min	HPLC, ^1^H NMR and LC-MS	0.73	[23]
	Hp, AcO, W		50 °C, 10 min	LC-ESI-MS, HPLC, ^1^H-NMR	0.90	
**UAE**	MeOH		25 °C, 3 h	HPLC	0.03	[6]
**Other Species**						
**HAE**	*Cladosiphon okamuranus*	MeOH	RT, 1 h	HPLC-DAD	0.27	[31]
	*Dictyota dicotoma*		RT, 24 h		6.42	[14]
	*Fucus distichus*		RT, 12 h		0.90	[32]
	*Saccharina japonica*		RT, 15 min		0.07	[33]
	*Saccharina sculpera*		RT, 12 h		0.70	[32]
	*Sargassum horneri*		RT, 12 h		3.70	[32]
	*Sargassum thunbergii*		RT, 12 h		1.80	[32]
**SAE**	*Feldmannia mitchelliae*	EA	80 °C, 16 h	HPLC	5.50	[19]
	*Saccharina japonica*	n-Hx	40 °C, 16 h	HPLC-DAD	0.45	[33]
	*Sargassum swartzii* C. Agardh	EA	80 °C, 6 h	FT-IR, ^1^H-NMR, ^13^C-NMR	0.17	[34]
**MAE**	*Laminaria japonica*	Hp, AcO, W	50 °C, 10 min	LC-ESI-MS, HPLC, ^1^H-NMR	0.04	[23]
	*Sargassum fusiforme*				0.02	[23]
**UAE**	*Padina tetrastromatica*	EtOH	50 Hz, 30 min	HPLC-DAD	0.75	[35]

*Extraction techniques*: Heat-assisted extraction (HAE); Soxhlet-assisted extraction (SAE); Microwave-assisted extraction (MAE); Ultrasound-assisted extraction (UAE). *Solvent*: Ethanol (EtOH); Ethyl Acetate (EA); Heptane (Hp); Hexane (Hex); Acetone (AcO); Water (W); Methanol (MeOH). *Extraction conditions*: Room temperature (RT)*. Detection method*: High-Performance Liquid Chromatography (HPLC); HPLC with Evaporative Light Scattering Detector (HPLC–ELSD); HPLC with Diode Array Detector (HPLC-DAD); Proton Nuclear Magnetic Resonance (^1^H NMR); Carbon-13 Nuclear Magnetic Resonance (^13^C NMR); Liquid Chromatography Mass Spectrometry (LC-MS); Liquid Chromatography-Electrospray Ionization coupled to Mass Spectrometry (LC-ESI-MS); Fourier Transform-Infrared Spectroscopy (FT-IR); Ultraviolet-Visible (UV). *: Fresh Wight.

**Table 2 marinedrugs-21-00282-t002:** Experimental design for the MAE and UAE extraction from *Undaria pinnatifida*. Experimental RSM results of the *CCCD* for the optimization of the three main variables involved (*X*_1_, *X*_2_, and *X*_3_) in the MAE and UAE extraction of Fx from *Undaria pinnatifida* for the three studied responses (*Y_1_, Y_2_,* and *Y_2/_Y_1_*). The experimental domain and codification of independent variables in the *CCCD* factorial design with five range levels are also shown.

	*Coded Values*			*Natural Values*						*Experimental Responses*					
				*MAE*			*UAE*			*MAE*			*UAE*		
	*X* _1_	*X* _2_	*X* _3_	*X*_1_: *t*	*X*_2_: *P*	*X*_3_: *S*	*X*_1_: *t*	*X*_2_: *Pw*	*X*_3_: *S*	*Y* _1_	*Y* _2_	*Y_2_/Y_1_*	*Y* _1_	*Y* _2_	*Y_2_/Y_1_*
				min	Bar	%	min	W	%	%	(mg Fx/g A)	(mg Fx/g E)	%	(mg Fx/g A)	(mg Fx/g E)
**1**	−1	−1	−1	*7.1*	*4*	*36.2*	*15.1*	*181.1*	*48.2*	37.17	0.05	0.13	9.55	0.80	8.40
**2**	−1	−1	1	*7.1*	*4*	*83.8*	*15.1*	*181.1*	*86.8*	35.77	8.83	24.70	8.93	3.64	40.73
**3**	−1	1	−1	*7.1*	*10*	*36.2*	*15.1*	*418.9*	*48.2*	50.63	0.01	0.02	45.79	7.75	16.93
**4**	−1	1	1	*7.1*	*10*	*83.8*	*15.1*	*418.9*	*86.8*	43.68	2.18	5.00	32.88	15.66	47.61
**5**	1	−1	−1	*18.9*	*4*	*36.2*	*44.9*	*181.1*	*48.2*	50.42	0.03	0.05	9.14	0.91	9.94
**6**	1	−1	1	*18.9*	*4*	*83.8*	*44.9*	*181.1*	*86.8*	39.48	6.74	17.08	13.29	5.39	40.54
**7**	1	1	−1	*18.9*	*10*	*36.2*	*44.9*	*418.9*	*48.2*	51.22	0.00	0.00	30.24	4.16	13.76
**8**	1	1	1	*18.9*	*10*	*83.8*	*44.9*	*418.9*	*86.8*	41.62	0.70	1.68	23.05	12.53	54.36
**9**	1.68	0	0	*23*	*7*	*60*	*5.0*	*300.0*	*67.5*	49.07	0.89	1.82	20.05	7.02	35.03
**10**	−1.68	0	0	*3*	*7*	*60*	*55.0*	*300.0*	*67.5*	24.75	5.40	21.81	34.05	15.94	46.80
**11**	0	−1.68	0	*13*	*2*	*60*	*30.0*	*100.0*	*67.5*	43.53	5.79	13.31	9.80	3.70	37.77
**12**	0	1.68	0	*13*	*12*	*60*	*30.0*	*500.0*	*67.5*	52.55	0.18	0.34	24.38	16.15	66.26
**13**	0	0	−1.68	*13*	*7*	*20*	*30.0*	*300.0*	*35.0*	55.29	0.00	0.00	52.05	0.09	0.17
**14**	0	0	1.68	*13*	*7*	*100*	*30.0*	*300.0*	*100.0*	16.24	1.62	10.00	11.14	13.68	122.85
**15**	−1.68	−1.68	−1.68	*3*	*2*	*20*	*5.0*	*100.0*	*35.0*	45.43	0.58	1.27	13.94	0.08	0.57
**16**	−1.68	−1.68	1.68	*3*	*2*	*100*	*5.0*	*100.0*	*100.0*	13.61	9.53	70.04	5.47	6.79	124.07
**17**	−1.68	1.68	−1.68	*3*	*12*	*20*	*5.0*	*500.0*	*35.0*	55.91	0.00	0.00	47.46	0.55	1.16
**18**	−1.68	1.68	1.68	*3*	*12*	*100*	*5.0*	*500.0*	*100.0*	14.90	2.29	15.34	8.86	10.03	113.11
**19**	1.68	−1.68	−1.68	*23*	*2*	*20*	*55.0*	*100.0*	*35.0*	52.08	0.41	0.79	33.52	0.08	0.24
**20**	1.68	−1.68	1.68	*23*	*2*	*100*	*55.0*	*100.0*	*100.0*	15.14	10.18	67.27	5.70	9.20	161.48
**21**	1.68	1.68	−1.68	*23*	*12*	*20*	*55.0*	*500.0*	*35.0*	58.43	0.00	0.00	45.33	0.48	1.07
**22**	1.68	1.68	1.68	*23*	*12*	*100*	*55.0*	*500.0*	*100.0*	29.82	0.00	0.00	15.09	20.89	138.49
**23**	0	0	0	*13*	*7*	*60*	*30.0*	*300.0*	*67.5*	47.37	4.88	10.30	43.65	20.36	46.65
**24**	0	0	0	*13*	*7*	*60*	*30.0*	*300.0*	*67.5*	48.03	5.04	10.49	45.40	15.64	34.46
**25**	0	0	0	*13*	*7*	*60*	*30.0*	*300.0*	*67.5*	50.80	5.14	10.12	51.55	18.46	35.81
**26**	0	0	0	*13*	*7*	*60*	*30.0*	*300.0*	*67.5*	51.58	5.13	9.94	39.50	18.46	46.73
**27**	0	0	0	*13*	*7*	*60*	*30.0*	*300.0*	*67.5*	50.64	5.00	9.87	46.69	20.52	43.95
**28**	0	0	0	*13*	*7*	*60*	*30.0*	*300.0*	*67.5*	50.04	5.13	10.26	45.66	17.48	38.29

The italics represent the main variables and the not italic numbers represent the experimental results obtained.

**Table 3 marinedrugs-21-00282-t003:** Divided into two parts. ***A:*** Parametric results of the third-order polynomial equation of Equation (8) for the three studied responses (*Y*_1_, *Y*_2_, and *Y*_2_/*Y*_1_), according to the CCCD with five range levels (Table 2). The parametric subscript 1, 2, and 3 stand for the variables involved *t* (*X*_1_), *P* or *Pw* (*X*_2_), and *S* (*X*_3_), respectively. The analysis of the significance of the parameters (*α* = 0.05) is presented in coded values. Additionally, the statistical information of the fitting procedure to the model is presented. ***B:*** Optimal conditions of the variables in natural values that lead to optimal response values for RSM using a CCCD for each response and technique.

		*MAE*						*UAE*					
		*Y* _1_		*Y* _2_		*Y_2_/Y_1_*		*Y* _1_		*Y* _2_		*Y_2_/Y_1_*	
		%		(mg Fx/g A)		(mg Fx/g E)		%		(mg Fx/g A)		(mg Fx/g E)	
**A: PARAMETRIC AND STATISTICAL ANALYSIS**													
Intercept	*b_0_*	49.87	±1.68	4.26	±0.12	11.00	±0.50	45.41	±2.41	18.49	±1.33	39.12	±5.83
Linear Effect	*b_1_*	--		−0.31	±0.06	−2.09	±0.35	−4.47	±3.28	--	--	--	
	*b_2_*	2.61	±0.81	−1.38	±0.06	−6.54	±0.35	15.24	±3.28	4.76	±1.81	--	
	*b_3_*	--		2.79	±0.20	6.54	±0.35	--		3.40	±0.54	--	
Quadratic Effect	*b_11_*	−3.87	±1.31	--		--		−6.49	±1.70	−2.48	±0.94	−5.36	±4.54
	*b_22_*	--		−0.69	±0.10	--		−10.01	±1.70	−3.03	±0.94	--	
	*b_33_*	−4.27	±1.31	−0.79	±0.10	--		−4.88	±1.70	−4.10	±0.94	8.05	±4.54
Cubic Effect	*b_111_*	1.08	±0.32	--		--		1.79	±1.29	--		--	
	*b_222_*	--		--		--		−3.86	±1.29	−1.09	±0.71	--	
	*b_333_*	−3.73	±0.32	−0.56	±0.08	--		−3.04	±0.39	--		13.01	±1.11
Interactive Linear Effect	*b_12_*	--		--		--		-1.03	±0.70	--		--	
	*b_13_*	--		--		--		--		0.55	±0.38	2.21	±2.00
	*b_23_*	--		−0.77	±0.04	−5.20	±0.25	−1.61	±0.70	0.68	±0.38	--	
	*b_123_*	0.48	±0.36	--		--		0.70	±0.43	--		--	
Interactive Quadratic Effect	*b_1122_*	--		--		--		−6.26	±4.73	−4.91	±2.61	--	
	*b_1133_*	--		--		--		--		--		--	
	*b_2233_*	--		0.36	±0.05	0.95	±0.12	--		--		--	
	*b_112233_*	0.39	±0.21	--		--		3.85	±1.48	2.39	±0.82	0.95	±0.74
**Statistics (*R²*)**		0.9426		0.8752		0.8695		0.9395		0.9034		0.9225	
**B: OPTIMAL VARIABLE CONDITIONS**													
*X_1_: t* (min)		10.27	±1.60	3.00	±0.87	3.00	±0.87	21.63	±2.33	30.46	±2.76	35.16	±2.96
*X_2_: P* (bar) or *Pw* (W)		12.00	±1.73	2.00	±0.71	2.00	±0.71	388.68	±9.86	374.46	±9.68	300.00	±8.66
*X_3_: S* (%)		20.00	±2.24	100.00	±5.00	100.00	±5.00	35.00	±2.96	76.55	±4.37	100.00	±5.00
**Optimal Response**		60.25	±4.99	10.01	±0.77	58.83	±3.86	54.13	±4.57	20.91	±1.66	124.39	±2.18

## Data Availability

Not applicable.

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
