# Peer review of "Emerging Technologies to Extract Fucoxanthin from Undaria pinnatifida: Microwave vs. Ultrasound Assisted Extractions"

_marinedrugs, 2023, doi:10.3390/md21050282_

Round 1

Reviewer 1 Report

C. Lourenço-Lopes et al . reports about emerging technologies to extract fucoxanthin from Undaria 2 pinnatifida: Microwave vs Ultrasound assisted extractions. After close evaluation of the manuscript I suggest revision of the manuscript according to the next points.

1. Please explain abbreviations in the abstract.

2, In Introduction: Ecologically friendly extraction of Fx using natural deep eutectic solvents was recently reported for brown seaweeds. Please update

3. Fig. 3A - both 3D and 2D plots represents the same data. Please consider to decrease the number of figures..

4. In Sect 3.1: the reproductive phase is important for the yield of active compounds from seaweeds (see for example https://doi.org/10.1007/s10811-022-02885-x). Please indicate at which reproductive phase the seaweeds were collected.

5. The conclusion is too long, The first paragraph of conclusion is reliable for introduction and aim of the study, rather for conclusion.

6. The second paragraph of conclusion is too long and not focused. It seems as a part of discussion.

7. The limitations of study were not mentioned.

8. Please explain how the results of the study can be used in industry. Can these results be scaled up?

Author Response

  1. Please explain abbreviations in the abstract.

Answer: Thank you for your suggestion. The abbreviations have been included in the abstract.

L23: In this regard, the present study aims to optimize the extraction conditions to obtain the highest Fx yield from U. pinnatifida through emerging techniques, namely microwave-assisted extraction (MAE) and ultrasound-assisted extraction (UAE); and compare them with the conventional methodologies of heat-assisted extraction (HAE) and Soxhlet-assisted extraction (SAE).

  1. In Introduction: Ecologically friendly extraction of Fx using natural deep eutectic solvents was recently reported for brown seaweeds. Please update.

Answer: Thank you for your comments. Some information has been added about the use of deep eutectic solvents on brown algae.

L86: Recently, deep eutectic solvents (DESs) have been reported for the extraction of Fx from brown microalgae (Tisochrysis lutea) [12] and macroalgae (Fucus vesiculosus) [13]. DESs have desirable properties such as thermal stability, adjustable viscosity, polarity, and high solubilization strength to extract compounds like Fx and advantages such as biodegradability, low toxicity and cost, easy production, and ecologically friendly with the environment. However, their application is still limited [12,13].

  1. Fig. 3A - both 3D and 2D plots represents the same data. Please consider decreasing the number of figures.

Answer: Many thanks for your suggestions. The authors intended this figure to give an example of how to interpret the data. Some people may have difficulties understanding 3D plots, so the idea (only in this figure 3) was to illustrate both plots with an example. Nevertheless, if you still consider that it will be better to delete 2D plots, we will perform the related modifications. Please feel free to give us your feedback on this aspect.

  1. In Sect 3.1: the reproductive phase is important for the yield of active compounds from seaweeds (see for example https://doi.org/10.1007/s10811-022-02885-x). Please indicate at which reproductive phase the seaweeds were collected.

Answer: Thank you for your suggestions. We have included the reproductive phase of the algae (mature sporophyte) and the season when the algae were collected.

L422: The alga was manually collected from Galician coastlines in December 2019 (mature sporophytes) and provided by the company Algamar (www.algamar.com).

  1. The conclusion is too long, The first paragraph of conclusion is reliable for introduction and aim of the study, rather for conclusion.

Answer: Thank you for your suggestion. This paragraph has been reduced and the conclusions have been revised.

L554: Macroalgae are valuable marine resources due to their diverse secondary metabolites like phenolic compounds and pigments for developing new ingredients or fortifying products in the food and cosmetic industries. Fucoxanthin (Fx) is one of the major carotenoids found in brown algae and its structure is responsible for their diverse properties, like strong antioxidant, anti-cancer, anti-inflammatory and anti-obesity activities, among others. The optimization of the extractive process of this molecule is of great interest to the industry due to the several therapeutic activities described.

SAE, the reference method for the extraction of Fx, obtained an extraction yield of up to 168.83 mg E/g A dw. The 2 h extraction obtained 3.68 mg Fx/g A dw and 21.90 mg Fx/g E dw whereas the 4 h extraction obtained higher values of 4.58 mg Fx/g A dw and 30.80 mg Fx/g E dw. After the optimization of the UAE and MAE extraction, we can conclude that even though the yield was a bit higher for MAE (60.25% against 54.13%), UAE was able to obtain higher concentrations of Fx (20.91 mg Fx/g A for UAE and 10.01 mg Fx/g A for MAE) and increase the purity of the extracts (124.39 mg Fx/g E for UAE and 58.83 mg Fx/g E for MAE). In general, MAE obtained better results for shorter times, low pressure, and higher concentration of ethanol whereas UAE needed medium times, medium power, and higher ethanol concentration. These results are far higher than the ones previously described in the literature and higher than the ones obtained with the conventional SAE. Both methodologies have the potential for industrial scaling up allowing for the obtention of higher yields, and minimizing the solvent and energy used thanks to their short extraction times. The study limitations are related to the selected ranges of the studied variables and the industrial scaling requirements of these techniques. Further research should focus on the isolation and purification of Fx after extraction.

  1. The second paragraph of conclusion is too long and not focused. It seems as a part of discussion.

Answer: Thank you for your suggestion. This paragraph has been deleted from the conclusion.

  1. The limitations of study were not mentioned.

Answer: Thank you for your suggestion. The main limitations of the study have been included in the last part of the conclusions along with future research expectations.

The study limitations are related to the selected ranges of the studied variables and the industrial scaling requirements of these techniques. Further research should focus on the isolation and purification of Fx after extraction.

  1. Please explain how the results of the study can be used in industry. Can these results be scaled up?

Answer: Thank you for your question. From our point of view, these results can be scaled up in the industry. The authors have provided mathematical equations to predict the extraction yield and fucoxanthin concentration that allows the industry to play with the variables and combine these mathematical expressions with others that consider the economic and environmental aspects, among others.

Author Response

1- All abbreviations in the text should be specified when they first appear.

Answer: Thank you for your suggestion. The abbreviations have been included and revised along the text.

2- Introduction: please explain more about the ultrasound and microwave assisted extraction methods and their mechanism. How these techniques improve extraction yield?

Answer: Thank you for your suggestion. The authors have included more information about UAE and MAE mechanisms and how they improve the extraction yield.

L103: This technique combines solvent extraction with microwave heating power. The energy is transmitted as waves, penetrating the matrix, and interacting with polar molecules, generating heat that increases the kinetics of the extraction. The cell structure is disrupted, and the solute is dissolved into the solvent, diffusing out of the matrix [20,21]. The conversion of electromagnetic energy into calorific energy takes place thanks to two simultaneous mechanisms: ionic conduction and dipole rotation [21].

L127: This technique employs ultrasonic waves with frequencies between 20 kHz and 10 MHz, between audible waves and microwave ranges [25]. It is based on the physicochemical principle of acoustic cavitation, that consists in the formation, growth, and collapse of bubbles present in a solvent-induced by ultrasonic waves. The propagation of ultrasonic waves through the solvent involves the formation of intermittent regions of high and low pressures, that generate gas bubbles. These bubbles grow and lead to their compression and rarefaction (expansion), reaching a critical size prior to their collapse [26,27].

3- Figure 2: HPLC results should be compared with their standards. Also, the image quality is not good (fig 2, P1-P7).

Answer: Thank you for your suggestion. HPLC results were compared according to the calibration curves presented in section 3.3. Chemical analysis through HPLC-DAD (L458-465) and discussed in section 2.1. Pigment identification in moderate conditions (L185-195). Regarding the image quality, the authors regret this inconvenience. The image is 600 ppp; however, when transforming the file to pdf, the quality is lost.

4- How is possible that ultrasound extraction for 2 and 4 hr without any fx degradation? Also please describe results in depth.

Answer: Thank you for your suggestion. In our study, we tested Fx to 78.4ºC and we found that no degradation occurred during 4 h. Moreover, the authors have developed previous studies where U. pinnatifida was subjected to high temperatures and long times without Fx degradation (5-65ºC and 30- 9680 min) (Lourenço-Lopes et al., 2022). In fact, this study concluded that temperatures around 45ºC, and times > 1200 min were recommended.

Lourenço-Lopes, C., Fraga-Corral, M., Soria-Lopez, A., Nuñes-Estevez, B., Barral-Martinez, M., Silva, A., Li, N., Liu, C., Simal-Gandara, J., & Prieto, M. A. (2022). Fucoxanthin’s Optimization from Undaria pinnatifida Using Conventional Heat Extraction, Bioactivity Assays and In Silico Studies. Antioxidants, 11(7), 1296. https://doi.org/10.3390/antiox11071296.

In this study, Fx was subjected to 78.4°C (EtOH ebullition temperature) and resisted degradation during 4 h extraction. The extraction was performed in duplicate with two selected times (2 and 4 h). Although the extraction yield was almost unchanged, between 168.30 and 168.83 mg E/g A dw, some differences were found regarding the specific pigment content of the obtained extracts.

5- Line 205-206: (Previous studies with SAE obtained much lower yields with 0.05 mg/g dw from U. pinnatifida using EtOH as solvent for 12 h [16].) why the yield was lower than this manuscript while the time was higher?

Answer: Thank you for your question. As the authors explained in their article, the results might be due to some contribution from the physical properties of U. pinnatifida (0.5–3 cm directly fed without pre-treatment). Bigger particles present lower ratios of surface area to volume. Also, there was the possibility of fucoxanthin degradation in ethanol during Soxhlet extraction (12 h). In our case, we tested that up for 4 h, and there was no degradation, but higher times might lead to worse yield results.

6- “2.3.1. Variable selection for the experimental design” section: line 227-232. Please insert related reference. Also, line 241-243: (EtOH was selected and the studied ethanol percentage (X3) from 35 to 100% for UAE; and 20 to 100% for MAE was chosen.) for obtain the best comparison, the same conditions should be used. Why the percentages of EtOH were different?

Answer: Thank you for your question. In previous unpublished results in our laboratory, we found that, according to the chemical structure of fucoxanthin and the different energies used for MAE and UAE, fucoxanthin appeared from 35-100% EtOH in UAE and 20-100% in MAE. Given these previous findings, we decided to focus the experimental design on those ranges where fucoxanthin appears without considering those lower percentages (0-35% in UAE and 0-20 in MAE) where fucoxanthin is not properly extracted.

7- Line 248: vegetable matrixes: algae is not vegetable.

Answer: Thank you for your suggestion. The text was misleading, it has been corrected to make it clearer that algae are not vegetables.

These two methods have been proven to improve the extraction of compounds from vegetable matrixes, and Fx from other brown algae.

8- Line 263-267: (From Table 2, higher yields of extraction (Y1) were obtained in the case of MAE, ranging from 13 to 56% whereas 5-48% for UAE. In both cases, the highest results were obtained for the experimental run number 17, using shorter times, maximum energies (power or pressure) and lesser percentages of EtOH concentrations. However, for the concentration of Fx (Y2), higher extractions were obtained for UAE than MAE, corresponding up to 20.89 and 10.18 mg Fx/g A, respectively.) please describe clearly, how the yield was higher in MAE while, Fx concentration was higher in UAE?

Answer: Thank you for your suggestion. We have noticed a mistake since the best experimental run for Y1 was different from MAE and UAE (17 and 13, respectively) and the same for Y2/Y1 ratio (16 and 20, respectively). On the other hand, the difference in the yields might be related to the energy applied by each technique, MAE obtained slightly higher yields (in terms of dw) while UAE was able to selectively extract more Fx.

From Table 2, higher yields of extraction (Y1) were obtained in the case of MAE, ranging from 13 to 56% whereas 5-52% for UAE. For MAE, the highest result was obtained for experimental run number 17, using shorter times, maximum energy, and a lesser percentage of EtOH concentration. For UAE, run number 13 obtained the highest yield using medium time and energy, and a low percentage of EtOH concentration. However, for the concentration of Fx (Y2), higher extraction yields were obtained for UAE than MAE, corresponding up to 20.89 and 10.18 mg Fx/g A, respectively. This difference might be related to the energy applied by each technique, MAE obtained slightly higher yields (in terms of dw) while UAE was able to selectively extract more Fx. Considering the Y2/Y1 ratio, it was maximum for the experimental run number 16 for MAE, using shorter time, minimum energy, and the highest EtOH concentration, obtaining 67.27 mg Fx/g E. For UAE, run number 20, with minimum energy, the longest time, and the highest EtOH concentration obtained 161.48 mg Fx/g E. This first approximation, without fitting any model, already gives an idea about the efficiency of UAE over MAE.

9- Methods should be described in details and authors should mention the references they used. Also, please check the order of methods. For example, usually first extraction is took place and then yield is measured.

Answer: Thank you for your suggestion. The methods were revised: the extraction yield was moved forward (3.5.) and supplementary references were added.

López, C. J., Caleja, C., Prieto, M. A., Barreiro, M. F., Barros, L., & Ferreira, I. C. F. R. (2018). Optimization and comparison of heat and ultrasound assisted extraction techniques to obtain anthocyanin compounds from Arbutus unedo L. Fruits. Food Chemistry, 264, 81–91. https://doi.org/10.1016/j.foodchem.2018.04.103

Lourenço-Lopes, C., Fraga-Corral, M., Garcia-Perez, P., Carreira-Casais, A., Silva, A., Simal-Gandara, J., & Prieto, M. A. (2022). A HPLC‐DAD method for identifying and estimating the content of fucoxanthin, β‐carotene and chlorophyll a in brown algal extracts. Food Chemistry Advances, 1, 100095. https://doi.org/10.1016/J.FOCHA.2022.100095

Pinela, J., Prieto, M. A., Carvalho, A. M., Barreiro, M. F., Oliveira, M. B. P. P., Barros, L., & Ferreira, I. C. F. R. (2016). Microwave-assisted extraction of phenolic acids and flavonoids and production of antioxidant ingredients from tomato: A nutraceutical-oriented optimization study. Separation and Purification Technology, 164, 114–124. https://doi.org/10.1016/j.seppur.2016.03.030.

10- This article is about fucoxanthin extraction, while there was no further purification of fucoxanthin from crud extract. Indeed, the fucoxanthin content was measured in different extracts obtained from different methods. The title may be need to change.

Answer: Thank you for your suggestion. With the title, we do not intend to say that fucoxanthin was purified, but indeed it was extracted with two technologies, MAE and UAE from U. pinnatifida. We would like to keep the title if it is correct for you.

Emerging technologies to extract fucoxanthin from Undaria pinnatifida: Microwave vs. Ultrasound assisted extractions

- English can be improved.

Answer: Thank you for your suggestion. The complete text has been revised.

Round 2

Reviewer 1 Report

Authors have revised the manuscriptm however some my comments were not addressed completely

1.  Sect 3.1: the reproductive phase is important for the yield of active compounds from seaweeds (see for example https://doi.org/10.1007/s10811-022-02885-x). Please discuss this question.

2. Tha phrase "The authors have provided mathematical equations to  predict the extraction yield and fucoxanthin concentration " is not totally correct. The mathematical equations  were calculated based on limited diapasons (see Table 2 -Experimental design). Please modify.

Author Response

Authors have revised the manuscript however some my comments were not addressed completely.

1. Sect 3.1: the reproductive phase is important for the yield of active compounds from seaweeds (see for example https://doi.org/10.1007/s10811-022-02885-x). Please discuss this question.

Answer: Thank you for your suggestions. We are sorry if we did not understand well previously. We included the reproductive phase of the algae (mature sporophyte) and the season when the algae were collected. Now we have included a brief sentence explaining the importance of the reproductive phase and provided the reference as an example.

L446: The alga was manually collected from Galician coastlines in December 2019 (mature sporophytes) and provided by the company Algamar (www.algamar.com). The reproductive phase and season should be considered when analyzing and discussing the obtained data, due to their influence on the yield of active compounds [51].

2. The phrase "The authors have provided mathematical equations to predict the extraction yield and fucoxanthin concentration " is not totally correct. The mathematical equations were calculated based on limited diapasons (see Table 2 -Experimental design). Please modify.

Answer: Thank you for your correction. We have modified the text accordingly. Please see below:

L435: Mathematical equations were calculated based on limited diapasons (Table 2) to predict the extraction yield and fucoxanthin concentration that can be combined with other equations defined by the industry that consider the economic and environmental aspects, among others.

Reviewer 2 Report

The manuscript is about optimization of fucoxanthin extract by ultrasound and microwave assisted extraction from Undaria pinnatifida, however, needs minor revision according to the comment:

·       Please recheck the order of methods. Because, the extract from conventional method was used for HPLC identification. Therefore, “Chemical analysis through HPLC-DAD” section might be better after extraction methods and also, mention that only the extract from conventional method used for identification.

Author Response

The manuscript is about optimization of fucoxanthin extract by ultrasound and microwave assisted extraction from Undaria pinnatifida, however, needs minor revision according to the comment:

Please recheck the order of methods. Because, the extract from conventional method was used for HPLC identification. Therefore, “Chemical analysis through HPLC-DAD” section might be better after extraction methods and also, mention that only the extract from conventional method used for identification.

Answer: Thank you for your suggestion. The authors have modified the order of the methods according to your suggestions:

3.1. Samples collection

3.2. Conventional heat-assisted extraction for pigment identification

3.3. Soxhlet-assisted extraction as a reference method

3.4. Chemical analysis through HPLC-DAD

3.5. Extraction yield

3.6. Optimization of the extraction process from Undaria pinnatifida

Answer: Also, it was mentioned that only the extract from conventional method used for identification:

The pigments’ content of the extract obtained by HAE was determined by a Waters HPLC equipment [including Waters 600 controller and Waters 600 pump, Waters 2996 photodiode array detector (DAD) (1.2 nm optical resolution), Waters 717 plus autosampler, and an AF in-line degasser from Waters].

Round 3

Reviewer 1 Report

The manuscript was revised according to ,y recommendations. The manuscript could be accepted in present form.